# MULTI-ENVIRONMENT TOPIC MODELS

## ABSTRACT

Probabilistic topic models are a powerful tool for extracting latent themes from large text datasets. In many text datasets, we also observe per-document covariates (e.g., source, style, political affiliation) that act as environments that modulate a "global" (environment-agnostic) topic representation. Accurately learning these representations is important for prediction on new documents in unseen environments and for estimating the causal effect of topics on real-world outcomes. To this end, we introduce the Multi-environment Topic Model (MTM), an unsupervised probabilistic model that separates global and environment-specific terms. Through experimentation on various political content, from ads to tweets and speeches, we show that the MTM produces interpretable global topics with distinct environment-specific words. On multi-environment data, the MTM outperforms strong baselines in and out-of-distribution. It also enables the discovery of accurate causal effects.[1]

## 1 INTRODUCTION

Topic models are a powerful tool for text analysis, offering a principled and efficient method for extracting latent themes from large text corpora. These models have wide-ranging applications in text representation and in causal analysis (Blei et al., 2003; Blei and Lafferty, 2006; Sridhar et al., 2022; Roberts et al., 2014).

Many text corpora include per-document covariates such as source, ideology, or style, which influence how the topics are represented Rosen-Zvi et al. (2012); Roberts et al. (2014). These covariates can be thought of as per-document "environmental" factors that modulate the global topics. Learning representations of topics while accounting for per-environment variations is particularly important when predicting new documents with unseen covariate configurations or performing causal analyses.

Table 1: Top terms learned by the MTM for a *U.S military* topic learned from political ads in Republican and Democrat-leaning regions in the U.S. Topic words related to the U.S military, such as '*veterans*' and '*troops*', receive high probability across all regions. However, top values in Republican-leaning regions show that words like '*freedom*' and '*terrorists*' receive high probability. In contrast, terms like '*home*' are more likely in ads from Democrat-leaning regions.

| Source | Top Words |
|---|---|
| Global | *america*, *veterans*, *war*, *proud*, *iraq*, *military*, *troops* |
| Republican-leaning | *terror*, *liberties*, *isis*, *terrorism*, *freedom*, *terrorists*, *defeat* |
| Democrat-leaning | *iraq*, *stay*, *guard*, *veterans*, *soldiers*, *port*, *home* |

To illustrate, consider a collection of political advertisements from multiple U.S. news channels. While all channels discuss similar topics, such as the *U.S military*, the manner of discussion varies by channel. Topic models might mistakenly conflate topic and channel variations, learning separate military topics for each channel. This issue poses a problem in two main scenarios.

First, for a model to generalize to new unseen channels, topic distributions should reflect channel-independent themes. Failure to do so can result in spurious associations between channels and topics, leading to poor predictive performance (see Section 5) (Peters et al., 2016; Arjovsky et al., 2019). Second, when using topic proportions as variables in causal studies (as treatments or as confounders) (Ash and Hansen, 2023), covariates such as the chosen channel are pre-treatment variables that need

---

[1]We implement the MTM in anonymous GitHub repository.

be adjusted for. For instance, when studying the impact of political ads on election results (Ash et al., 2020), failing to adjust for channel variations may result in biased causal estimates (see Section 6).

To address these issues, we propose the Multi-environment Topic Model (MTM). The MTM is a hierarchical probabilistic model designed to analyze text from various environments, separating universal terms from environment-specific terms. The MTM assumes that the effect of an environment on the global topic distribution is sparse. That is, for each topic most words are shared across all environments, and only a subset are environment-specific. To enforce sparsity, we employ an automatic-relevance determination prior (ARD) (MacKay, 1992). We fit MTMs with auto-encoding variational Bayes (Kingma and Welling, 2013). Table 1 shows an example of what MTMs uncover.

Our contributions are as follows:

- We introduce the MTM, which captures consistent and interpretable topics from multiple environments.
- We create three datasets that facilitate the comparison of text models across different environments (ideology, source, and style), including held-out, out-of-distribution environments.
- We demonstrate that the MTM achieves lower perplexity in both in-distribution and out-of-distribution scenarios compared to strong baselines.
- We show that the MTM enables the discovery of true causal effects on multi-environment data.

Section 2 discusses related work on topic modeling, multi-environment learning and treatment discovery. Section 3 details the construction of the MTM. Section 4 explains how to infer topics using the MTM. Section 5 presents our empirical studies, which compare the MTM to strong baselines on multi-environment datasets. Section 6 demonstrates how existing topic models can lead to biased causal estimates and how the MTM mitigates this issue. Finally, Section 7 explores limitations and future directions for multi-environment probabilistic models.

## 2 RELATED WORK

**Topic Models.** Probabilistic topic models uncover latent themes in large datasets (Blei et al., 2003; Blei and Lafferty, 2009; Vayansky and Kumar, 2020). Topics uncovered with such models are commonly used for text analysis (Ash and Hansen, 2023) and for estimating causal effects with text data (Feder et al., 2022a). Many topic models incorporate per-document covariates to learn topics that are predictive of certain outcomes or that reflect different data-generating processes (Rosen-Zvi et al., 2012; Roberts et al., 2014; Sridhar et al., 2022). Other topic models incorporate environment specific information, such as the Structural Topic Model (STM) (Roberts et al., 2016) and SCHOLAR (Card and Smith, 2018). These models differ from MTMs in that they assume covariates influence topic proportions. In contrast, the MTM posits that while the same topics are discussed across environments, they are framed differently, focusing on environment-specific variations in word usage rather than shifts in topic proportions. Sparse priors are commonly used in Bayesian models for enhancing interpretability, and are often complemented by empirical Bayes (EB) methods for parameter estimation (Tipping, 2001). Carvalho et al. (2010) combine these approaches to develop the horseshoe estimator, and Brown and Griffin (2010) to study normal-gamma priors. Efron (2012) provides a comprehensive overview of EB methods. Building on this literature, we use the ARD prior, which relies on a gamma distribution with parameters learned via EB (MacKay, 1992). While topics models like KATE (Chen and Zaki, 2017) and the Tree-Structured Neural Topic Model (Isonuma et al., 2020) enforce sparsity on the topic-word distribution, the MTM applies sparsity to environment-specific deviations of the topic-word distribution.

**Invariant learning from multiple environments.** Invariant learning tackles the problem of learning models that generalize across different environments. Invariant learning through feature pruning was pioneered by Peters et al. (2016), and has since been developed for variable selection (Magliacane et al., 2018; Heinze-Deml et al., 2018) and representation learning (Arjovsky et al., 2019; Wald et al., 2021; Puli et al., 2022; Makar et al., 2022; Jiang and Veitch, 2022). These methods have been applied in a range of domains, including in natural language processing (Veitch et al., 2021; Feder et al., 2021; 2022b; Zheng et al., 2023; Feder et al., 2024). For causal estimation, invariant learning ensures stable representations by accounting for confounding variables (Shi et al., 2021; Yin et al., 2021). Our work considers a related problem of learning stable representations of text from multiple environments, focusing on a probabilistic approach.

**Topics as treatments in causal experiments.** A common approach to studying the effects of text is treatment discovery, which involves producing interpretable features of text that can be causally linked to outcomes (Feder et al., 2022a). Probabilistic topics models are interpretable and can be trained without direct supervision, making them the preferred method of choice for social scientists in these settings Grimmer et al. (2021); Ash and Hansen (2023). For example, Fong and Grimmer (2016) discovered features of candidate biographies that drive voter evaluations, and Hansen et al. (2018) estimated a topic model based on the transcripts of the Federal Open Market Committee. Several recent papers have also applied latent Dirichlet allocation (Blei et al., 2003) to newspaper corpora and interpreted the content of topics in terms of economic phenomena (Mueller and Rauh, 2018; Larsen and Thorsrud, 2019; Thorsrud, 2020; Bybee et al., 2021). Our experiments contribute to this literature (Section 6) by demonstrating the importance of using a topic model that is faithful to the true data-generating process (multi-environment data in our case) when using topics as textual treatments in a causal study.

## 3 MULTI-ENVIRONMENT TOPIC MODELS

Consider a corpus of $n$ text documents with the corresponding environment-specific information represented as $\mathcal{D} = \{(\mathbf{w}_1, \mathbf{x}_1), \ldots, (\mathbf{w}_n, \mathbf{x}_n)\}$, where each document $\mathbf{w}_i$ is paired with its corresponding feature vector $\mathbf{x}_i$. Each document $\mathbf{w}_i$ is a sequence of $m$ word tokens, given by $\mathbf{w}_i = \{w_{i1}, \ldots, w_{im}\}$, that come from a vocabulary $w_{ij} \in \mathbb{1}^{|V|}$. The feature vectors $\{\mathbf{x}_1, \ldots, \mathbf{x}_n\}$ capture the environment-specific information associated with each document in the corpus. For each document, the environment is represented by $\mathbf{x}_i \in \{0, 1\}^{|E|}$. $\mathbf{x}_i$ could be an indicator vector that represents the channel each advertisement in a dataset emerged from, or more generally represent the political affiliation (Republican or Democrat) or style (speech, article, or tweet) of a document.

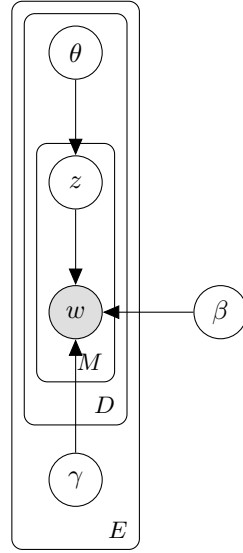

Figure 1: A graphical model for the multi-environment topic model (MTM). $M$ denotes words in a document and $D$ documents. $E$ denotes the environments documents are drawn from (determined by different configurations of covariates $\mathbf{x}$). $z$ denotes topic assignment, $\beta$ denotes global weights for each word in the vocabulary, and $\gamma$ denotes environment-specific weights.

Our goal is to learn global topics and their per-environment adjustments. Recall our running example, where news outlets discuss the same topics in unique ways. We want to capture the unique ways these outlets discuss the same topic while simultaneously extracting common terms shared among all outlets.

In topic modeling, each document is represented as a mixture of topics, with a local latent variable $\theta_i$ denoting the per-document topic intensities. Topics are denoted by $\beta$, and each $\beta_k$ is a probability distribution over the vocabulary, $\beta_k \in \mathbb{R}^v$. We introduce a new latent variable, $\gamma_k \in \mathbb{R}^{e \times v}$, where $k \in \{1, \ldots, K\}$, that is designed to capture the effect that each environment has on each topic-word distribution, $\beta_k$. In a multi-environment topic model, each document is assumed to have been generated through the following process:

1. Draw $\beta_k \sim \mathcal{N}(\cdot, \cdot)$, $\beta_k \in \mathbb{R}^v$, $k = 1, \ldots, K$.
2. Draw $\gamma_k \sim p(\gamma)$, $\gamma_k \in \mathbb{R}^{e \times v}$, $k = 1, \ldots, K$.
3. For each document $i$:
    (a) Draw topic intensity $\theta_i \sim \mathcal{N}(\cdot, \cdot)$.
    (b) For each word $j$:
        i. Choose a topic assignment $z_{ij} \sim \text{Cat}(\pi(\theta_i))$.
        ii. Choose a word $w_{ij} \sim \text{Cat}(\pi(\beta_z + \gamma_z \cdot x_i))$

The graphical model for the multi-environment topic model is represented in Figure 1. Given data, the posterior finds the topic and word distributions that best explain the corpus, and the distribution that best explain the words that are most probable in each environment. For example, given advertisements

displayed in Republican-leaning and Democrat-leaning regions of the U.S, the posterior for MTM uncovers the topic of *U.S military* as shown in Table 1. MTM represents the terms discussed by both outlets in $\beta_k$, while capturing the particular ways Republican and Democrat-leaning channels discuss military action $\gamma_k$. We next specify $p(\gamma)$ using the automatic relevance determination (ARD) prior.

**Automatic Relevance Determination (ARD) and Empirical Bayes.** MTMs are built with the additional assumption that documents are generated based on different configurations of observed covariates. The goal is to separate global from environment-specific information. To do this, we introduce a new latent variable, $\gamma_k$. We further posit that environment effects on the global topic-word distribution $\beta_k$ should be sparse. Consider again our running example of ads from Republican and Democrat-leaning sources. In this case, nearly all words will be shared across sources, so we want to ensure $\gamma_k$ only places high density on terms that are highly probable for a particular source.

In many real-world tasks, the input data contains a large number of irrelevant features. ARD is a method used to filter them out (MacKay, 1992; Tipping, 2001). Its basis is to assign independent Gaussian priors to the feature weights. Given the feature weights $\eta$, the ARD assigns priors as:

$$\sigma_c \sim \text{Gamma}(a, b) \tag{1}$$

$$p(\eta|\alpha) = \prod_c \mathcal{N}(\eta_c|0, \alpha_c^{-1}). \tag{2}$$

The precisions, $\alpha = \{\alpha_c\}$, represent a vector of hyperparameters. Each hyperparameter $\alpha_i$ controls how far its corresponding weight $\eta_c$ is allowed to deviate from zero. Rather than fixing them a-priori, ARD hyperparameters are learned from the data by maximizing the the likelihood of the data with empirical Bayes (Carlin and Louis, 2000; Efron, 2012).

In the MTM, ARD places the prior of $\gamma_{e,k,v}$:

$$\sigma_{e,k,v} \sim \text{Gamma}(a, b)$$

$$\gamma_{e,k,v} \sim \mathcal{N}(0, \sigma_{e,k,v}^{-1}).$$

We set the parameters of the Gamma distribution by maximizing the likelihood of the data:

$$\hat{a}, \hat{b} = \arg\max_{a,b} p(\mathcal{D}|a, b). \tag{3}$$

This prior encourages the majority of the environment-specific deviations to exhibit strong shrinkage. It drives them towards zero, while allowing some to possess significant non-zero values. We incorporate it into the MTM to highlight influential environment-specific effects ($\gamma$), while still allowing $\beta$ to capture most of the variation across documents. In Section 5 we discuss the importance of this modeling choice.

## 4 INFERENCE

With the MTM defined, we now turn our attention to procedures for inference and parameter estimation. MTMs rely on multiple latent variables: topic-word distributions $\beta$, document-topic proportion $\theta$, and environment-specific deviations on the topic-word distribution $\gamma$. Conditional on the text and document specific features, we perform inference on these latents through the posterior distribution $p(\theta, z, \beta, \gamma|\mathcal{D})$, where $\mathcal{D} = \{(\mathbf{w}_1, \mathbf{x}_1), \ldots, (\mathbf{w}_n, \mathbf{x}_n)\}$.

As calculating this posterior is intractable, we rely on approximate inference. We use black-box variational inference (BBVI) Ranganath et al. (2014). Using the reparameterization trick we marginalize out $z_{ij}$, leaving us with only continuous variables (Kingma and Welling, 2013).

We rely on mean-field variational inference to approximate the posterior distribution (Jordan et al., 1999; Blei et al., 2017). We set $\phi = (\theta, \beta, \gamma)$ as the variational parameters, and let $q_\phi(\theta, \beta, \gamma)$ be the family of approximate posterior distribution, indexed by the variational parameters. Variational inference aims to find the setting of $\phi$ that minimizes the KL divergence between $q_\phi$ and the posterior (Blei et al., 2017). To approximate $\theta$, we use an encoder neural network that takes $\mathbf{w}_i$ as input and consists of one hidden layer with 50 units, ReLU activation, and batch normalization. Minimizing this KL divergence is equivalent to maximizing the evidence lower bound (ELBO):

$$\text{ELBO} = \mathbb{E}_{q_\phi}[\log p(\theta, \beta, \gamma) + \log p(x|\theta, \beta, \gamma) - \log q_\phi(\theta, \beta, \gamma)]. \tag{4}$$

To approximate the posterior, we use the mean-field variational family, which results in our latent variables, $\theta$, $\beta$, and $\gamma$ being mutually independent and each governed by a distinct factor in the variational density. We employ Gaussian factors as our variational densities, thus our objective is to optimize the ELBO with respect to the variational parameters:

$$\phi = \{\mu_\theta, \sigma_\theta^2, \mu_\beta, \sigma_\beta^2, \mu_\gamma, \sigma_\gamma^2\}.$$

The model parameters are optimized using minibatch stochastic gradient descent in PyTorch by minimizing the negative ELBO. To achieve this optimization, we employ the Adam optimizer (Kingma and Ba, 2014). The complete algorithm is described in Algorithm 1.

## 5 EMPIRICAL STUDIES

Our empirical studies are driven by five questions:

1. How stable is the perplexity of MTMs when tested on datasets from different environments?
2. How does stability change when we incorporate environment-specific information ($\gamma$) when calculating MTMs' perplexity?
3. How does MTMs' performance compare to other topic model variants?
4. How does using a non-sparse prior on $\gamma$ effect model performance?
5. In situations where using $\theta$ as text representations produce biased causal estimates, can the MTM lead to more accurate estimates of causal effects?

We find that:

1. In all test settings, the predictive power of the MTM is stable across environments, especially when incorporating environment-specific effects ($\gamma$).
2. When using environment-specific effects from an irrelevant environment (i.e., using article-specific effects to calculate perplexity for speeches), perplexity drops considerably.
3. Compared to baselines, the MTM has better perplexity on in and out-of-distribution data.
4. Using a non-sparse prior on $\gamma$ results in significant decrease in performance.
5. Using the topic proportions from the MTM allows uncovering accurate causal effects.

### 5.1 BASELINES

We compare the multi-environment topic model to the relevant baselines:

- **LDA** - Latent Dirichlet allocation (LDA) represents a variant of online Variational Bayes inference for learning (Blei et al., 2003; 2017).
- **Vanilla Topic Model** - The vanilla topic model (VTM) represents the base version of our model without any environment-specific variations:

$$\theta_i \sim \mathcal{N}(\cdot, \cdot)$$
$$\beta_k \sim \mathcal{N}(\cdot, \cdot)$$
$$w_{ij} \sim (\pi(\theta_i)\pi(\beta)).$$

- **Non-sparse MTM** - The non-sparse multi-environment (nMTM) represents the MTM, but with a Normal distribution on the $\gamma$ prior.
- **ProdLDA** - ProdLDA represents the distribution over individual words has a product of experts rather than the mixture model used in LDA (Srivastava and Sutton, 2017; Hinton, 2002). We use the standard implementation in Pyro (Bingham et al., 2018).
- **MTM + $\gamma$** - The MTM + $\gamma$ represents the sum of the environment specific effects from a particular environment, $\gamma_{k,v}$, to $\beta_{k,v}$. We want to evaluate how the performance shifts when including environment-specific information. For example, in Figure 2 MTM + $\gamma$ represents the perplexity when using $\gamma_{k,v} + \beta_{k,v}$, rather than solely $\beta_{k,v}$, where the $\gamma_{k,v}$ is the learned article specific effects on the global topic-distribution, $\beta$.
- **BERTopic** - BERTopic generates topics by clustering document embeddings from pre-trained transformer models, and uses TF-IDF to identify the top words in each cluster. Topic proportions are calculated by by comparing documents to each document cluster.
- **SCHOLAR** - SCHOLAR represents a neural topic model that builds on the Structural Topic Model (Roberts et al., 2014) by using variational autoencoders. It integrates environment-specific information, allowing the model to flexibly adjust topic distributions (Card and Smith, 2018). Without environment information it defaults to ProdLDA.

## 5.2 EVALUATION METRICS

We evaluate topic models using perplexity, topic coherence, and causal estimation. While perplexity is an imperfect measure of topic models Chang et al. (2009), it remains useful for assessing topic stability across environments and evaluating the generalizability of models to data from different distributions. Similarly, like perplexity, automated topic coherence metrics have known limitations (Hoyle et al., 2021). Coherence is often leveraged for exploratory data analysis Chang et al. (2009). Our focus is on using model parameters for causal estimation settings, where researchers use topic models to discover interpretable features of text that can be causally linked to outcomes. For this reason, we also evaluate the models based on their ability to produce accurate causal effect estimates.

## 5.3 DATASETS

To empirically study the MTM and the baselines, we construct 3 multi-environment datasets.

**Ideological Dataset.** The ideological dataset consists of US political advertisements from the last twenty years. We split the dataset by ideology, and have an even amount of advertisements from Republican and Democrat politicians ($12,941$ samples each). We test all models on three held-out datasets: Republican-only politicians, Democrat-only politicians, and an even mixture of advertisements from both parties.

**Style Dataset.** The style dataset consists of news articles, senator tweets, and senate speeches related to U.S. immigration. The U.S. immigration articles are gathered from the Media Framing Corpus (Card et al., 2015). We use all $4,052$ articles in the dataset. We augment the dataset used by Vafa et al. (2020), which is based on an open-source set of tweets of U.S. legislators from 2009–2017. We create a list of keywords related to immigration and sample $4,052$ tweets that contain at least one of the keywords; we repeat the process for Senate speeches from the 111-114th Congress. (Gentzkow et al., 2018). The environments for the style dataset are defined by the distinct writing styles of tweets, speeches, and articles.

**Channels dataset.** The channels dataset consists of political advertisements run on TV channels across the United States. We create our two environments by splitting the original dataset and assigning channels from Republican voting regions to one environment, and channels from Democratic voting regions to the other. Appendix C.1 presents the characteristics of each dataset.

## 5.4 IN-DISTRIBUTION PERFORMANCE

We compare the perplexity and NPMI of the MTM against baseline models using held-out data from the same sources observed during training. The number of topics is set to $k = 20$, and all probabilistic models are trained for 150 epochs. All results are averaged across three runs. We note that when we have test data from a distribution that is unseen during training we do not have access to environment-specific $\gamma$s. Thus, we can not use $\gamma$ when calculating perplexity for out-of-distribution test data. For all subsequent analyses, even when our test data is from the same distribution as our training data, we evaluate the MTM without $\gamma$.

Figure 2 compares performance across models on the **ideological** dataset, which has two environments, represented by Republican and Democrat political advertisements. We train on an even number of ads from each environment. Figure 2 represents perplexity of our baseline models and the MTM. We see in Figure 2 that the MTM performs significantly better on all test sets. Even when using only the global topic distribution, $\beta$, perplexity is stable on both test sets.

When using Republican-leaning ideological effects in the perplexity calculation for Republican advertisements we have better perplexity than using $\beta$ only; however, when we use Republican-leaning effects on the Democrat-leaning test set performance declines considerably. This indicates that the information captured in $\gamma$ is relevant to a specific environment, Republican ads, while uninformative to Democrat ads. The non-sparse MTM variant performs worse in relation to the MTM with the ARD prior, conveying the importance of employing a sparse prior. We visualize the top terms that $\gamma_k$ places high density on in Table 15.

Figure 3 represents perplexity of LDA, ProdLDA, SCHOLAR, the VTM, nMTM, and MTM when trained on the **channels** dataset. The MTM satisfies our desiderata: its predictive performance is

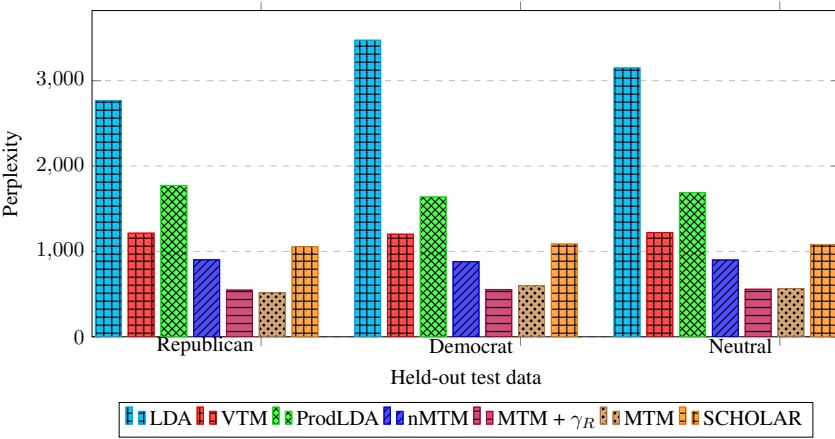

Figure 2: Perplexity on held-out data across models trained on the **ideological** dataset, consisting of political advertisements from Republican and Democrat politicians. The MTM $+\gamma_R$ represents global $\beta$ with Republican-specific deviations $\gamma_R$. MTM outperforms all baselines on all three test sets.

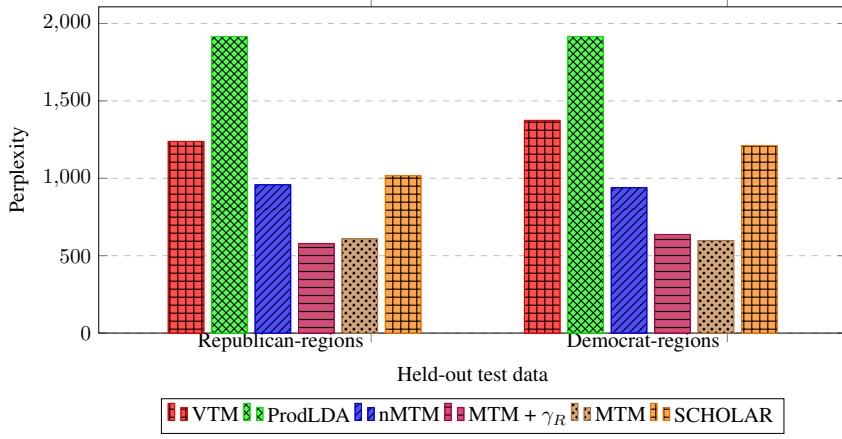

Figure 3: Perplexity on held-out data across models trained on a dataset of political advertisements from **channels** across different regions of the U.S. The MTM $+\gamma_R$ represents global $\beta$ with Republican-specific deviations $\gamma_R$. MTM outperforms all baselines across all regions.

consistent across environments, performance declines when using environment-specific effects $\gamma_k$ from an environment that differs from the environment of a test set, and perplexity is better than baseline models.

We find that the MTM performs slightly worse than some baseline models on NPMI, as shown in Table 2. Achieving a high NPMI score depends on the top words from each topic co-occurring frequently within the same document. However, these co-occurring words are not necessarily shared across all environments, meaning that a model with a high NPMI score can still conflate global and environment-specific words. In Section 6.2, we demonstrate that models with higher NPMI scores, such as LDA, provide less accurate causal estimates compared to the MTM.

## 5.5 OUT-OF-DISTRIBUTION PERFORMANCE

We investigate how the MTM fits data from unseen (out-of-distribution) environments using the **style** dataset, which contains three environments: articles, tweets, and speeches. Table 3 shows the perplexity of the VTM, ProdLDA, nMTM, and MTM when trained on speeches and articles and tested on tweets. We do not included SCHOLAR

Table 2: NPMI on the **ideology** and **channels** datasets.

| Model | Ideology | Channels |
|---|---|---|
| MTM | $-0.16$ | $-0.1$ |
| VTM | $-0.13$ | $-0.12$ |
| BERTopic | $-0.27$ | $-0.22$ |
| LDA | $-0.13$ | $\mathbf{-0.8}$ |
| ProdLDA | $\mathbf{-0.10}$ | $-0.14$ |

as a baseline because tweets were not seen during training.
MTM performs better than baselines on the in-distribution
tests. On out-of-distribution data the performance gap increases. Further, we find that the nMTM
performs worse than the MTM and VTM, highlighting the importance of sparse priors on $\gamma$. Using
the same training data, we also test on political advertisements and find again the MTM outperforms
the VTM. Table 18 in Appendix D presents the full results.

We combine environments from the **ideological** dataset with the **style** dataset to train on political ads and articles, and test on tweets. Table 4 presents the perplexity of our baseline models and the MTM. We find that the MTM outperforms the baselines on both the in-distribution and out-of-distribution tests. We find the sparse prior on $\gamma$ to be an important factor in improving model robustness. Without sparsity, MTMs capture too much global information in $\gamma$ (Table 21), hurting out-of-distribution performance. Implementation details are described in Appendix C.

Table 3: Performance (held-out perplexity) across environments when training on congressional senate speeches and news articles. The MTM has substantially lower perplexity, especially when tested on the out-of-distribution tweets.

| Model | In-Distribution | | OOD |
| | Articles | Speeches | Tweets |
| --- | --- | --- | --- |
| VTM | $1,613$ | $1,598$ | $2,206$ |
| ProdLDA | $5,162$ | $2,406$ | $13,807$ |
| nMTM | $2,030$ | $1,987$ | $2,143$ |
| MTM | **1,502** | **1,524** | **1,690** |

Table 4: Performance (held-out perplexity) across environments when training on political ads and news articles. The MTM has substantially lower perplexity, especially when tested on the out-of-distribution tweets.

| Model | In-Distribution | | OOD |
| | Articles | Ads | Tweets |
| --- | --- | --- | --- |
| VTM | $1,689$ | $1,159$ | $1,793$ |
| ProdLDA | $2,293$ | $2,698$ | $9,454$ |
| nMTM | $1,841$ | $1,468$ | $1,757$ |
| MTM | **1,254** | **662** | **1,221** |

# 6 CAUSAL INFERENCE WITH TOPIC PROPORTIONS

In the social sciences, learned topic proportions ($\theta$s) are often used as a low-dimensional interpretable representation of text in causal studies. We describe here the problem with using topic proportions from an unadjusted topic model (like the VTM) to represent text in a causal study on multi-environment data (Section 6.1). We then demonstrate empirically why the MTM is crucial for this setup (Section 6.2).

## 6.1 WHY ADJUST FOR ENVIRONMENT COVARIATES?

Consider a dataset $\mathcal{D} = \{(\mathbf{w}_i, \mathbf{x}_i, y_i)\}_{i=1}^n$, where $\mathbf{w}_i$ are words in document $i$, $\mathbf{x}_i$ are pre-treatment measurements (i.e. the channel that the ad will run on), and $y_i$ is the outcome variable. Imagine we are interested in estimating the causal effect of a topic $T_i$ chosen for document $i$ (e.g., exposure to a specific topic $k$) on the outcome $y_i$. The treatment $T_i$ is some measure based on the topic proportions $\theta_i$ (e.g., a binary indicator for whether topic $k$ received the most amount of mass) (Ash et al., 2020).

In the potential outcomes framework (Rubin, 1974), we denote $y_i(T_i)$ as the potential outcome for unit $i$ under treatment $T_i$. The average treatment effect (ATE), controlling for pre-treatment variables $X$, is defined as:

$$\tau = \mathbb{E}[y_i(T_i = 1) \mid \mathbf{x}_i] - \mathbb{E}[y_i(T_i = 0) \mid \mathbf{x}_i],$$

where $T_i = \mathbf{1}\{\arg\max_j \theta_{ij} = k\}$.

A confounder is a variable that influences both the treatment and the outcome. In our context, $\mathbf{x}_i$ are covariates that affect the outcome $y_i$ (e.g., choosing which channel to run the ad on causally affects voting outcomes), and might be baked into the topic proportions $\theta_i$ in a topic model (e.g. when topics include channel information). If we do not adjust for $\mathbf{x}_i$ when learning $\theta_i$, our estimate of the treatment effect might be biased.

Denote the true topic proportions as $\theta_i$. When $\mathbf{x}_i$ affects topic assignment, the learned topic proportions $\hat{\theta}_i$ will be given by: $\hat{\theta}_i = f(\theta_i, \mathbf{x}_i)$. Outcome $y_i$ is influenced by both the topic $T_i$ and the

confounders $\mathbf{x}_i$: $y_i = g(T_i) + h(\mathbf{x}_i) + \eta_i$, where $g(\cdot)$ is the effect of the chosen topic, $h(\cdot)$ is the effect of the covariates, and $\epsilon_i$ is the error term.

Substituting this into the causal effect estimation, we get:

$$\hat{\tau} = \mathbb{E}[y_i \mid \mathbf{1}\left\{\arg\max_j f(\theta_{ij}, \mathbf{x}_i) = k\right\}, \mathbf{x}_i] - \mathbb{E}[y_i \mid \mathbf{1}\left\{\arg\max_j f(\theta_{ij}, \mathbf{x}_i) \neq k\right\}, \mathbf{x}_i].$$

Comparatively, the true causal effect $\tau$ is:

$$\tau = \mathbb{E}[y_i \mid \mathbf{1}\left\{\arg\max_j \theta_{ij} = k\right\}, \mathbf{x}_i] - \mathbb{E}[y_i \mid \mathbf{1}\left\{\arg\max_j \theta_{ij} \neq k\right\}, \mathbf{x}_i].$$

In any case where $\hat{\theta}_i$ is not conditionally independent of $\mathbf{x}_i$ (as in the VTM), we will get that $\hat{\tau} \neq \tau$. By modeling $\hat{\theta}_i^{MTM}$ as the sum of $\beta_i$ and $\gamma_{k,x_i}$, the MTM controls for variation in $x_i$ and ensures that $\theta_i$ is conditionally independent of $x_i$. We now turn to empirically test the efficacy of using topic proportions from MTM and the baseline models for causal estimation on semi-synthetic data.

## 6.2 ESTIMATING CAUSAL EFFECTS OF TOPICS

Table 5: The top terms for the topic distributions related to energy for the MTM, VTM, and LDA models, which were trained on the **ideological** dataset. The VTM identifies two distinct topics associated with energy-related discourse, each reflecting terminology predominantly used by either Democrat or Republican viewpoints. LDA identifies a topic related to energy, but it also reflects Republican viewpoints. For the MTM, variations in word association across political ideologies are captured through the $\gamma$ parameter, and it successfully learns a single topic for energy.

| Model | Source | Top Words |
|---|---|---|
| MTM | $\beta_k$: Global | *energy, oil, choice, world, gas, prices, power, broken, coal, faith* |
| | $\gamma_k$: Republican | *kill, coal, ballot, keystone, faith, destroy, domestic, face, epa, broken* |
| | $\gamma_k$: Democrat | *oil, gouging, clean, price, climate, renewable, alternative, wind, progress, nextgen* |
| VTM | $\beta_k$ (Topic 15) | *tax, money, dollars, values, energy, breaks, sales, corporations, spend, increase, gas, reform* |
| | $\beta_k$ (Topic 21) | *america, fight, oil, gas, world, fought, billions, foreign, military, states, coal, freedom* |
| LDA | $\beta_k$ | *oil, energy, gas, america, white, companies, foreign, drilling, progress, independence* |

Table 6: The top terms for the topic distributions related to senior social policies discovered by the MTM model on the **ideological** dataset.

| Source | Top Words |
|---|---|
| $\beta_k$: Global | *health, security, medicare, social, seniors, insurance, costs, drug, healthcare, companies* |
| $\gamma_k$: Republican | *takeover, bureaucrats, doctors, health, billion, choices, plans, canceled, sky-rocketing, log* |
| $\gamma_k$: Democrat | *companies, privatize, conditions, protections, insurance, health, social, voted, aarp, age* |

Based on the **ideological** dataset, we design two semi-synthetic experiments where we sample an outcome variable $Y$ from a Bernoulli distribution with parameter $p = 0.5$. First, we train the MTM, LDA, VTM, ProdLDA, and BERTopic on the **ideological** dataset with $k = 30$ and extract the topic proportions. We then model $Y$ as a function of a binary predictor $T$, where $T = 1$ if the topic proportion for either the '*energy*' or 'senior social policies' topic (as shown in Table 5 and Table 6) is the highest among all topic proportions in a given document, and $T = 0$ otherwise. To any ad containing two keywords from the energy list [*energy, oil, gas, clean*] or two from the senior social

policies list [*health*, *social*, *security*, *insurance*, *seniors*, *healthcare*, *pension*, *retirement*], we add $0.2$ to the outcome variable $Y$. We sample 700 ads for each list and additional 700, resulting in $2,100$ samples. We run separate ordinary least squares (OLS) regressions for each experiment.

We select the topic proportion corresponding to the $\beta_k$ that has the most overlapping words with the *energy* and *senior social policies* topics. Since VTM learns two separate topics with equal overlap with the energy keyword list, we fit a model where $T = 1$ if the combined topic proportions of the two energy topics (Topics 15 and 21) are the highest among all topics for a given document. To estimate the causal effect of the topics, we use the following OLS regression: $Y = \delta_0 + \delta_1 T + \delta_2 X + \epsilon$ where $\delta_1$ represents the marginal effect of the *energy* topic on $Y$ in one experiment, and the *senior social policies* topic in the other. The regression results from the experiments using the MTM, VTM, ProdLDA, LDA and BERTopic models are summarized in Table 7. We exclude SCHOLAR from our experiments because its modeling approach allows topic distributions to be influenced by environment-specific deviations, which contradicts our goal of obtaining global topic representations. However, we include ProdLDA, which is equivalent to SCHOLAR without environment-specific information. Using the representation from the MTM, we are able to capture the true effect of $0.2$ more accurately than any other model. Models such as LDA, which have higher coherence scores than MTM, perform worse when estimating causal effects. This is because assigning high probability to environment-specific terms can improve coherence metrics, but high coherence does not guarantee unbiased causal estimates when data comes from multiple environments. Table 5 shows how VTM and LDA can assign high probability to terms reflecting right-leaning ideology, such as 'military' and 'freedom', within the context of energy, whereas MTM effectively separates global topics from environment-specific effects. By separating global topics from environment-specific deviations, MTM controls for the confounding effects of environments, leading to more accurate causal estimates in the presence of data from different environments. Top words from all models are shown in Appendix D.3.

Table 7: The coefficient $\delta_1$ from the OLS regressions using various models for the '*energy*' and '*senior social policies*' topics. With MTM, we are able to learn substantial effects for both topics, while other models provide mixed results. Baseline models have the propensity to misrepresent the underlying topics when trained on data from multiple environments while MTMs are able to learn the environment-specific information in the $\gamma$ parameter and capture the global information in $\beta$. *Note:* ***$p < 0.001$, **$p < 0.01$, *$p < 0.05$.

| Model | Energy | | Senior Social Policies | |
|---|---|---|---|---|
| | $\delta_1$ **Coefficient** | **Std. Error** | $\delta_1$ **Coefficient** | **Std. Error** |
| MTM | **0.200**\*\*\* | 0.028 | **0.203**\*\*\* | 0.027 |
| VTM | 0.066 | 0.041 | 0.000 | 0.000 |
| LDA | -0.263 | 0.140 | 0.149\*\*\* | 0.037 |
| ProdLDA | 0.150\*\*\* | 0.029 | 0.085\* | 0.041 |
| BERTopic | 0.341 | 0.410 | 0.116\*\*\* | 0.032 |

## 7 DISCUSSION AND LIMITATIONS

We addressed the problem of modeling text from multiple environments. To that end, we developed the multi-environment topic model (MTM), an unsupervised probabilistic model that learns a global topic distribution and adjusts for environment-specific variation. The MTM has stable perplexity across different environments. It captures meaningful information in the environment-specific latent variable, performs better in and out of distribution and allows discovery of accurate causal effects.

The MTM has clear limitations, which opens up several avenues for future work. First, as MTMs rely on a bag-of-words representation, integrating them with more modern neural text representation models can potentially improve their predictive performance. Second, while we demonstrate that MTMs allow uncovering true causal effects in multi-environment data, we only evaluate this on semi-synthetic data. Exploring this question rigorously is out of the scope of this paper, but is an important problem to address in future work. Finally, another potential avenue for further exploration not addressed in this paper is the connection between invariant learning and probabilistic models.

**Reproducibility** We provide the code and data to use the MTM in an anonymous Github repo anonymous GitHub repository and also attach the code to our submission in a zipfile. Algorithm 1

also displays the algorithm for the MTM. In the Appendix C we include the MTM hyperparameters and tokenizer hyperparameters. Appendix C also includes a description of each dataset.

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

APPENDIX

# A  ALGORITHM

---
**Algorithm 1** Multi-environment topic model (MTM)

---
1: **Input:** Number of topics $K$, number of words $V$, number of environments $E$
2: **Output:** Document intensities $\hat{\theta}$, global topics $\hat{\beta}$, environment-specific effects on global topics $\hat{\gamma}$
3: **Initialize:** Variational parameters $\mu_\theta, \sigma_\theta^2, \mu_\beta, \sigma_\beta^2, \mu_\gamma, \sigma_\gamma^2$ randomly
4: **while** the *evidence lower bound* (ELBO) has not converged **do**
5:     sample a document index $d \in \{1, 2, \ldots, D\}$
6:     sample $z_\theta, z_\beta,$ and $z_\gamma \sim \mathcal{N}(0, I)$                              ▷ Sample noise distribution
7:     Set $\tilde{\theta} = \exp(z_\theta \odot \sigma_\theta + \mu_\theta)$                              ▷ Reparameterize
8:     Set $\tilde{\beta} = \exp(z_\beta \odot \sigma_\beta + \mu_\beta)$                              ▷ Reparameterize
9:     Set $\tilde{\gamma} = \exp(z_\gamma \odot \sigma_\gamma + \mu_\gamma)$                              ▷ Reparameterize
10:    **for** $v \in \{1, \ldots, V\}$ **do**
11:        Set $w_{dv} = \sum_k \tilde{\theta}_{dk}(\tilde{\beta}_{kv} + \tilde{\gamma}_{ekv})$                              ▷ Log-likelihood term
12:    **end for**
13:    Set $\log p(w_d|\tilde{\theta}, \tilde{\beta}, \tilde{\gamma}) = \sum_v \log p(w_{dv}|\tilde{\theta}, \tilde{\beta}, \tilde{\gamma})$                              ▷ Sum over words
14:    Compute $\log p(\tilde{\theta}, \tilde{\beta}, \tilde{\gamma})$ and $\log q(\tilde{\theta}, \tilde{\beta}, \tilde{\gamma})$                              ▷ Prior and entropy terms
15:    Set ELBO $= \log p(\tilde{\theta}, \tilde{\beta}, \tilde{\gamma}) + N \cdot \log p(w_d|\tilde{\theta}, \tilde{\beta}, \tilde{\gamma}) - \log q(\tilde{\theta}, \tilde{\beta}, \tilde{\gamma})$
16:    Compute gradients $\nabla_\phi$ELBO using automatic differentiation
17:    Update parameters $\phi$
18: **end while**
19: **Return** approximate posterior means $\hat{\theta}, \hat{\beta}, \hat{\gamma}$

---

# B  THE HORSESHOE PRIOR

Another way to enforce sparsity on the multi-environment topic model, is with a horseshoe prior for $\gamma$, which is defined as:

$$\gamma_{e,k,v} \mid \lambda_{ek}, \tau \sim \mathcal{N}(0, \lambda_{e,k}^2 \tau^2).$$

Here, $\lambda_{e,k}$ represents the local shrinkage parameter specific to each environment $e$ and topic $k$, while $\tau$ is the global shrinkage parameter that applies to all $\gamma$ variables. The horseshoe prior for $\lambda_{e,k}$ has the following characteristic form:

$$\lambda_{e,k} \sim \mathcal{C}^+(0, 1)$$
$$\tau \sim \mathcal{C}^+(0, 1)$$

where $\mathcal{C}^+(0, 1)$ denotes the standard half-Cauchy distribution, which has a probability density function that is flat around zero and has heavy tails. As such, the prior encourages the majority of these environment-specific deviations to exhibit strong shrinkage, driving them towards zero, while allowing some to possess significant non-zero values, thereby highlighting truly influential environment-specific effects and allowing $\beta$ to maintain its ability to capture topics across documents. Thus the hMTM disentangles global from environment-specific influences by capturing the global topics in $\beta$ and environment-specific deviations in $\gamma$.

In Table 8, words under the topic, $\beta_k$, related to the energy such as 'oil' and 'water' receive high density across all environments in a corpus which consists of political articles, tweets and senate speeches, whereas words such as 'projects' and 'infrastructure' receive high density in the $\gamma_k$ representing the senate speech-specific effects, and acronyms like 'EPA' receive high density in the twitter specific effects. Table 9 displays the top terms the hMTM learns in a topic related to healthcare when trained on the ideological dataset.

Table 8: The table displays the top words learned by hMTM when trained on the style dataset. The words in global topics appear in all environments when discussing a given topic, while the words that receive the top $\gamma_k$ values predominately appear in one environment. We observe distinctive word choices in tweets, articles, and senate speeches, reflecting different communication styles.

| Source | Top Words |
|---|---|
| $\beta_k$: Global | *energy, oil, water, jobs, air* 
 *card, credit, banks, financial, bank* |
| $\gamma_k$: News Articles | *canada, disaster, wind, property, construction* 
 *card, cards, fee, fraud, investment* |
| $\gamma_k$: Senate Speeches | *national, infrastructure, country, projects, climate* 
 *rules, consumers, industry, rates, regulatory* |
| $\gamma_k$: Tweets | *epa, climate, roll, environment, coal* 
 *competition, settlement, consumers, exchange, regulate* |

Table 9: When trained on the ideology dataset, hMTM learns interpretable environment-specific terms while simultaneously uncovering meaningful global topics.

| Source | Top Words |
|---|---|
| $\beta_k$: Global | *health, budget, debt, cost, costs* |
| $\gamma_k$: Republican | *takeover, debt, health, trillion, bureaucrats* |
| $\gamma_k$: Democrat | *health, affordable, healthcare, universal, medicaid* |

## C    EXPERIMENTAL DETAILS

### C.1    DATASETS

Table 10: A summary of the datasets we construct for testing topic models across multiple environments.

| Dataset | Style | Ideology | Political advertisements |
|---|---|---|---|
| **Focus of text Environments** | US Immigration {Tweets from US Senators, US Senate speeches, news articles} | Politics {Republican, Democrat} politicians | Politics Channels from {Republican, Democrat} voting regions |
| **Training set size** | $4,052$ per environment | $12,941$ per environment | $12,446$ per environment |

### C.2    STYLE DATASET

The style dataset consists of $12,156$ samples, with an even amount of samples from each environment. We constructed a vocabulary of unigrams that occurred in at least $0.6\%$ and in no more than $50\%$ of the documents. We use the same tokenization scheme for all baselines we compare to. We removed cities, states, and the names of politicians in addition to stopwords. For hMTM, we set $\lambda$ and $\tau$, parameters used in the horseshoe prior, to be $0.4$. For MTM, we set the hyperparameters of the gamma distribution, $a$ and $b$, to be $3.7$ and $0.34$ respectively. These values were determined by training our model for $50$ epochs, taking $2$ gradient steps for updating $a$ and $b$ in the empirical Bayes method for every $1$ step for the rest of the model. This approach helps guarantee that hyperparameter

updates are not overshadowed by the updates of the rest of the parameters in the model. We set the number of topics, $k$, to be 20 for all experiments in this paper.

**OOD experiments:**

When training on speeches and articles and testing on tweets the training dataset has 8104 samples. We constructed a vocabulary of unigrams that occurred in at least $0.8\%$ and in no more than $50\%$ of the documents. For MTM, we set the hyperparameters of the gamma distribution, $a$ and $b$, to be 2.92 and 0.25 respectively. These values were determined by training our model for 50 epochs, taking 2 gradient steps for updating $a$ and $b$ in the empirical Bayes method for every 1 step for the rest of the model.

When training on ads and articles and testing on tweets the training dataset has 8104 samples. We constructed a vocabulary of unigrams that occurred in at least $0.2\%$ and in no more than $50\%$ of the documents. For MTM, we set the hyperparameters of the gamma distribution, $a$ and $b$, to be 2.87 and 0.25 respectively. These values were determined by training our model for 50 epochs, taking 2 gradient steps for updating $a$ and $b$ in the empirical Bayes method for every 1 step for the rest of the model.

### C.3 IDEOLOGICAL DATASET

We construct a vocabulary of unigrams that occurred in at least $0.6\%$ and in no more than $40\%$ of the documents. We remov cities, states, and the names of politicians in addition to stopwords. For hMTM, we set $\lambda$ and $\tau$, parameters used in the horseshoe prior, to be 0.5. For MTM, we set the hyperparameters of the gamma distribution, $a$ and $b$, to be 4.0 and 0.11 respectively. These values were determined by training our model for 15 epochs, taking 2 gradient steps for updating $a$ and $b$ in the empirical bayes method for every 1 step for the rest of the model.

### C.4 POLITICAL ADS DATASET

The style dataset consists of $24,892$ samples, with an even amount of samples from each environment. We construct a vocabulary of unigrams that occurrs in at least $0.6\%$ and in no more than $40\%$ of the documents. We remove cities, states, and the names of politicians in addition to stopwords. For hMTM, we set $\lambda$ and $\tau$, parameters used in the horseshoe prior, to be 0.4. For MTM, we set the hyperparameters of the gamma distribution, $a$ and $b$, to be 3.8 and 0.13 respectively. These values were determined by training our model for 15 epochs, taking 2 gradient steps for updating $a$ and $b$ in the empirical Bayes method for every 1 step for the rest of the model.

### C.5 HYPERPARAMETERS

For auto-encoding VB inference, we used an encoder with two hidden layers of size 50, ReLU activation, and batch normalization after each layer. For stochastic optimization with Adam, we use automatic differentiation in PyTorch. We used a learning rate of 0.01 based on implementation from Sridhar et al. (2022). These methods were trained on a T4 GPU.

## D ADDITIONAL EXPERIMENTS AND RESULTS

### D.1 IN-DISTRIBUTION PERFORMANCE

**Channels dataset.**
Table 11 presents a sample advertisement from a Democrat and Republican-leaning region respectively.

Table 11: An example of advertisements from our dataset. KSWB is a San Diego based news channel, and WKRG is a station licensed to Mobile, Alabama.

| Source | Text |
|---|---|
| Alabama (WKRG) | *What does Governor Bob Riley call over 70,000 new jobs? A great start. His conservative leadership's given us the lowest unemployment in Alabama history, turning a record deficit into a record surplus. Now Governor Riley has delivered the most significant tax cuts in our history. The people get up every morning and work, they are the ones that allowed us to have the surplus. The only thing I'm saying, they should have some of it back. Governor Bob Riley, honest, conservative leadership.* |
| California (KSWB) | *State budget cuts are crippling my classroom. So I can't believe the Sacramento politicians cut a backroom deal that will give our state's wealthiest corporations a new billion dollar tax giveaway. A new handout that can only mean larger class sizes and even more teacher layoffs. But passing Prop 24 can change all that. Prop 24 repeals the unfair corporate giveaway and puts our priorities first. Vote yes on Prop 24 because it's time to give our schools a break, not the big corporations. their corporate giveaway and puts their priorities first. Vote yes on Prop 24 because it's time to give our schools a break, not the big corporations.* |

Table 12: Perplexity of the hMTM when trained on a dataset of political advertisements from channels in different regions of the U.S. $\gamma$ represents Republican leaning effects.

| Model | Republican | Democrat |
|---|---|---|
| hMTM + $\gamma$ | **545** | 664 |
| hMTM | 622 | **651** |

**Style dataset.**
Table 13 represents the perplexity of gensim LDA, vanilla topic model, ProdLDA, nMTM, and MTM. It also includes the performance when using environment-specific information, $\gamma$. Here $\gamma$ represents the article-specific effects on our topic-word distribution $\beta$. Notably, when using the article-specific effects for calculating perplexity on a test set consisting of only articles, the perplexity improves. Indicating that the article-specific effects captured in $\gamma$ uncover information relevant to articles. However, when we use article-specific effects to calculate the perplexity on speeches, the perplexity declines considerably, whereas when we use only $\beta$, our perplexity remains stable across test sets, indicating that it captures a robust distribution of topics. The non-sparse variant of the MTM, nMTM, performs worse than the MTM and also the VTM baseline, indicating the importance of placing a sparse prior on $\gamma$. We visualize the top terms that $\gamma_k$ places high density on in Table 14.

Table 13: Model perplexities when training on all three sources and testing on unseen data from each environment. $\gamma$ corresponds to article-specific effects. VTM, ProdLDA, and LDA are less stable than the MTM.

| Model | Articles | Speeches | Tweets |
|---|---|---|---|
| LDA | 9344 | 3007 | $3.936 \times 10^{12}$ |
| VTM | 1345 | 1461 | 1584 |
| ProdLDA | 2757 | 2427 | 2000 |
| nMTM | 1586 | 1754 | 1716 |
| hMTM | 1215 | 1306 | 1309 |
| hMTM + $\gamma$ | 1051 | 1333 | 1218 |
| MTM | 1181 | **1298** | 1112 |
| MTM + $\gamma$ | **1048** | 1426 | **1017** |

Table 14 reflects how MTM learns environment specific effects, and global topics when trained on the style dataset. In the topic related to immigration, $\beta$ captures words that are appear across environments like 'country' and 'law' whereas words like 'secretary' and 'homeland' are predominant in senate speeches and 'naturalization' is predominant in articles.

Table 14: Top words for a particular topic distribution learned by MTM when trained on the style dataset. The words in global topics appear across environments, while the words that receive the top $\gamma$ values predominantly appear in one environment. We observe distinctive word choices in tweets, articles, and senate speeches, reflecting different communication styles.

| Source | Top Words |
|---|---|
| $\beta_k$: Global Topics | *country*, *law*, *status*, *policy*, *illegal*, *immigrants*, *immigration*, *border*, *citizenship* |
| $\gamma_k$: News Articles | *immigration*, *primary*, *illegal*, *immigrants*, *legal*, *naturalization*, *states*, *driver*, *citizenship* |
| $\gamma_k$: Senate Speeches | *immigration*, *border*, *security*, *gang*, *secretary*, *everify*, *homeland*, *colleagues*, *america* |
| $\gamma_k$: Tweets | *country*, *discuss*, *policy*, *immigration*, *reform*, *illegal*, *applications*, *check*, *plan* |

**Ideological dataset.** Table 15 represents the top terms the MTM learns on ideological dataset. Table 16 reflects the perplexity of the hMTM across the different test sets.

Table 15: When trained on the ideological dataset MTM learns meaningful terms for the Republican and Democrat environments, while simultaneously uncovering meaningful global topics.

| Source | Top Words |
|---|---|
| $\beta_k$: Global | *health*, *seniors*, *insurance*, *medicare*, *plan*, *costs*, *drug*, *affordable*, *healthcare*, *fix* |
| $\gamma_k$: Republican | *obamacare*, *health*, *takeover*, *bureaucrats*, *replace*, *medicare*, *supports*, *repeal*, *lawsuits*, *choices* |
| $\gamma_k$: Democrat | *health*, *companies*, *protections*, *conditions*, *deny*, *insurance*, *prices*, *voted*, *drug*, *gut* |

Table 16: Perplexity performance of hMTM when trained on a dataset of political advertisements from Republican and Democrat politicians. hMTM with $\gamma$ represents a combination of the learned topic distribution $\beta$, where $\gamma$ indicates the Republican deviations on each word distribution of $\beta$.

| Model | Republican | Democrat | Neutral |
|---|---|---|---|
| hMTM | 547 | **541** | 551 |
| hMTM + $\gamma$ | **516** | 569 | **550** |

## D.2 OUT-OF-DISTRIBUTION PERFORMANCE

We further investigate how the MTM performs when tested on data that was unseen during training using our style dataset. We train on political news articles and senate speeches and then test on political advertisements. These political advertisements come from our ideological dataset.

Table 18 represents the perplexity of the VTM and MTM when trained on articles and speeches and tested on advertisements.

Table 17: hMTM also has lower perplexity than baseline models when tested on out-of-distribution data. Here we trained on congressional senate speeches and news articles and tested on tweets from U.S. senators.

| Model | Articles (Perplexity) | Speeches (Perplexity) | Tweets (Perplexity) |
|-------|------------------------|------------------------|----------------------|
| hMTM | $1,481$ | $1,451$ | $1,625$ |

Table 18: MTM has lower perplexity than baseline models when tested on out-of-distribution data. Here we trained on congressional senate speeches and news articles and tested on political advertisements.

| Model | Advertisements |
|-------|----------------|
| VTM | $1,771$ |
| ProdLDA | $8,912$ |
| nMTM | $2,131$ |
| MTM | $1,603$ |
| hMTM | **1,503** |

### D.3 ESTIMATING CAUSAL EFFECTS OF TOPICS

Table 19: The top terms for the topic distributions related to senior social policies for the MTM, VTM, ProdLDA, Gensim LDA, and BERTopic models.

| Model | Source | Top Words |
|-------|--------|-----------|
| MTM | $\beta_k$: Global | *health, security, medicare, social, seniors, insurance, costs, drug, healthcare, companies* |
|  | $\gamma_k$: Republican | *takeover, bureaucrats, doctors, health, billion, choices, plans, canceled, skyrocketing, log* |
|  | $\gamma_k$: Democrat | *companies, privatize, conditions, protections, insurance, health, social, voted, aarp, age* |
| ProdLDA | $\beta_k$ (Topic 21) | *security, medicare, social, seniors, protect, benefits, age, privatize, retirement, earned* |
| VTM | $\beta_k$ (Topic 0) | *health, medicare, seniors, insurance, costs, affordable, coverage, prescription, conditions, lower, drugs, cost, premiums, charge, deny* |
| Gensim LDA | $\beta_k$ (Topic 8) | *security, social, medicare, seniors, benefits, protect, cut, retirement, age, plan* |
| BERTopic | $\beta_k$ (Topic 2) | *health, social, medicare, insurance, security, planned, parenthood, seniors, drug, cancer* |

Table 20: The top terms for the topic distributions related to energy for the ProdLDA, LDA, and BERTopic models, which were trained on the **ideological** dataset.

| Model | Source | Top Words |
|---|---|---|
| ProdLDA | $\beta_k$ (Topic 1) | *energy, oil, clean, prices, gas, foreign, alternative, renewable, economy, drilling* |
| LDA | $\beta_k$ (Topic 10) | *oil, energy, gas, america, white, companies, foreign, drilling, progress, independence* |
| BERTopic | $\beta_k$ (Topic 19) | *oil, money, case, tied, fraud, illegal, outsider, ethics, interests, denounced* |

# E  HMTM VS MTM

Model criticism aims to identify the limitations of a model in a specific context and suggest areas for improvement (Blei, 2014; Gelman and Shalizi, 2012). Although hMTM and MTM exhibit strong performance compared to other topic model variants, it is crucial to verify the expected behavior of the newly introduced $\gamma$ parameter.

According to Occam's Razor principle, models with unnecessary complexity should not be preferred over simpler ones (MacKay, 1992). As indicated in Table 21, hMTM is less sparse and exhibits greater uncertainty regarding its parameter values compared to MTM. Employing the ARD prior leads to a $\gamma$ parameter that is not only more sparse but also more effective in capturing environment-specific terms. This is evident from MTM's superior performance on both in-distribution and out-of-distribution data. Besides having considerably lower perplexity, nMTM is also less sparse than both models.

We want to ensure that a given word $w$ that is highly probable in a certain environment $e_i$ and a specific topic $k$ occurs more frequently in documents discussing topic $k$ in environment $e_i$ than in documents discussing the same topic in a different environment $e_j$. We introduce a metric: count opposite. It represents the number of words (from the top 10 $\gamma$ words for each environment and each topic) that have a higher frequency in the test set environment opposite to the one they are associated with. For instance, if $\gamma$, in the context of a Republican-leaning environment, assigns a high probability to the word 'wasteful' occurring in discussions about taxation, this word should appear more frequently in a subset of Republican-leaning advertisements about taxation than in a subset of Democrat-leaning advertisements on the same topic. Among the words receiving high $\gamma$ values for a given environment and topic, these words are more likely to occur in the dataset corresponding to the environment represented by $\gamma$ in MTM than in hMTM for the same dataset. We find the median Count Opposite of the top 10 words for each topic and $\gamma$ environment is 1.0 for MTM and 2.0 for hMTM. Motivating the use of the ARD prior.

| Model | Group | Perp. | Sparsity | $\mu_\gamma$ | $\sigma_\gamma$ |
|---|---|---|---|---|---|
| nMTM | Republican | 949 | 3.6% | $7.1 \times 10^{-3}$ | 0.4 |
| | Democrat | 936 | 3.8% | $6.7 \times 10^{-3}$ | 0.4 |
| hMTM | Republican | 662 | 41.64% | $7.13 \times 10^{-4}$ | 0.21 |
| | Democrat | 651 | 42.70% | $-2.92 \times 10^{-3}$ | 0.24 |
| MTM | Republican | 598 | 79.95% | $5.45 \times 10^{-5}$ | 0.03 |
| | Democrat | 604 | 79.89% | $1.37 \times 10^{-4}$ | 0.03 |

Table 21: Comparing the sparsity of different variants of MTMs we find the MTM with an ARD prior to be the most sparse. Sparsity is defined as any value less than 0.01.

