# OpenReview forum: "Multi-environment Topic Models"
_ICLR.cc/2025/Conference — Submitted to ICLR 2025_

### Official Review · Reviewer_D1sR · 2024-10-28

**Soundness:** 2
**Presentation:** 3
**Contribution:** 3
**Rating:** 5
**Confidence:** 5

**Summary:**

This paper proposes the Multi-environment Topic Model (MTM), an unsupervised probabilistic model that separates global and environment-specific terms for the generated topics. This submission clearly states its differences from the existing methods that incorporate environment-specific information, including the Structural Topic Model (STM) and SCHOLAR. Besides, this study is well-motivated from the social science applications.

**Strengths:**

- This submission clearly states its differences from the existing methods that incorporate environment-specific information, including the Structural Topic Model (STM) and SCHOLAR. This makes it easier for the reader to capture the contribution of the proposed method.

- This study is well-motivated from the social science applications, with good experimental settings in Section 6.

**Weaknesses:**

- Appendices A, B, and E are not cited in the main content of this submission, which makes some of the content hard to follow.

- The number of topics is set to 20 for all experiments, which weakens the experimental evidence.

- Some parts of this submission are hard to understand, please refer to my questions.

**Questions:**

- Is there a typo in Table 2? I note that the NPMI of LDA is -0.8 over Channels, but it is marked in bold (which may indicate the best performance).

- In Section 3, should v and e be revised to |V| and |E|, respectively? If so, V and E in Algorithm 1 should also be respectively revised to |V| and |E|.

- In Equation 1, should $\sigma$ be revised to $\alpha$?

- In lines 286-287, the authors mention that they test all models on three held-out datasets, but I can not find the statistics of these held-out datasets. Can you provide the testing set size in Table 10?

- Is there a typo in “The style dataset” at the first sentence of Appendix C.4? Should it be revised to “The channel dataset” when referring to Appendix D.1 or “The political advertisements dataset” when referring to Table 10? I’m confused about distinguishing the last two datasets in Table 10.

- In lines 306-309, the authors mention that they have test data from a distribution that is unseen during training, which makes them not have access to environment-specific $\gamma$s. Besides, they cannot use $\gamma$ when calculating perplexity for out-of-distribution test data. Since I cannot find the corresponding steps in Algorithm 1, can you provide the details regarding the above strategy?

---

> ### Author Response · Authors · 2024-11-19
>
> >In Equation 1, should σbe revised to α?
>
> Yes, thank you for pointing this out!
>
>
> >In lines 286-287, the authors mention that they test all models on three held-out datasets, but I can not find the statistics of these held-out datasets. Can you provide the testing set size in Table 10?
>
> The test set is 10% of the training set size. We will incorporate this information in the paper.
>
> >Is there a typo in “The style dataset” at the first sentence of Appendix C.4? Should it be revised to “The channel dataset” when referring to Appendix D.1 or “The political advertisements dataset” when referring to Table 10? I’m confused about distinguishing the last two datasets in Table 10.
>
> Yes, thank you for pointing this out!
>
> >In lines 306-309, the authors mention that they have test data from a distribution that is unseen during training, which makes them not have access to environment-specific γs. Besides, they cannot use γ when calculating perplexity for out-of-distribution test data. Since I cannot find the corresponding steps in Algorithm 1, can you provide the details regarding the above strategy?
>
> We calculate the MTM performance with γ in none of the experiments, as stated in section 5.4. We only use beta when calculating perplexity. For MTM+γ, we use the combination of both.

---

> > ### Comment · Reviewer_D1sR · 2024-11-25
> >
> > Thanks for the authors' response and clarification.

---

### Official Review · Reviewer_TAie · 2024-11-01

**Soundness:** 3
**Presentation:** 3
**Contribution:** 2
**Rating:** 5
**Confidence:** 3

**Summary:**

This paper proposes a new multi-environment topic model (MTM) that learns a global
topic distribution and adjusts it for environment-specific variation.
The paper is generally well-written. The motivation is clear.
In the experiments, the perplexity results on held-out data across models show the superior performance of the proposed method.
However, there are some concerns with its effectiveness.

**Strengths:**

- The paper is generally well-written.
- The motivation is clear.
- Adjusting global topic distributions for environment-specific variation sounds interesting and feasible.
- In the experiments, the perplexity results on held-out data across models show the superior performance of the proposed method.

**Weaknesses:**

- The number of environments in the data used in the experiments  (just two or three) is rather small compared to the number of topics.
- The authors evaluate the causal effects in multi-environment only with the semi-synthetic data. The semi-synthetic data is conveniently designed for the proposed method to work as expected and better than other models that are not multi-environment aware.
- The proposed MTM performs slightly worse than some baseline models regarding NPMI.

**Questions:**

The number of environments in the data used in the experiments (just two or three) is rather small compared to the number of topics.
If the size of a dataset is sufficiently large, and each document in the dataset belongs to only one environment, then it can be divided into each environment-wise subset, and a topic model can be applied to each subset separately.
If they do so, they cannot obtain shared global topics across the dataset. But still, they may obtain better perplexity, higher topic coherence scores, and higher causal effects. I wonder how the proposed method scales with a larger number of environments.

The authors find that the proposed MTM performs slightly worse than some baseline models regarding NPMI and observe that this is because co-occurring words are not necessarily shared across all environments.
However, I wonder if the top words in the global topics are still supposed to be shared across all environments
because they are in the "global" topics. Does the environment-specific variation include "negative" variation to
counteract the global "top" words in some environments?

In Table 6, the most typical word "health" appears in all the Global, Republican, and Democrat.
Does this redundancy suggest that the inference is sub-optimal?

In Section 3, $E$ is not formally defined, and both $|E|$ and $e$ are used for the number of environments.

In Section 3, $\pi$   (softmax?) is used without its definition.

In Equation (1). $\sigma_c$ must be $\alpha_c$.

At the end of Page 6, Table 15 must be Table 5.

**Details Of Ethics Concerns:**

No ethics review is needed.

---

> ### Author Response · Authors · 2024-11-19
>
> > I wonder how the proposed method scales with a larger number of environments.
>
> Thank you for this suggestion. We focus on these types of datasets that are frequently used in the social science literature. Please let us know if you have any dataset suggestions.
>
> >In Table 6, the most typical word "health" appears in all the Global, Republican, and Democrat. Does this redundancy suggest that the inference is sub-optimal?
>
> It is possible that the inference is sub-optimal in some situations. In section E in the appendix we describe an experiment where we evaluate that a given word w that is highly probable in a certain environment ei and a specific topic k occurs more frequently in documents discussing topic k in environment ei than in documents discussing the same topic in a different environment ej. We find that the MTM does well at this.
>
> >Does the environment-specific variation include "negative" variation to counteract the global "top" words in some environments?
>
> Can you elaborate on what you mean? If our reading of this point is correct, we think that our model already does this. In our model, if a word does not appear in an environment, it is less likely to be considered in the global topic beta words.
>
> >However, I wonder if the top words in the global topics are still supposed to be shared across all environments because they are in the "global" topics.
>
> Yes they should be. Our point was to emphasize that it is completely possible for a baseline model that captures the environment specific variation in the top words for a topic to have a better NPMI score.
>
> >In Section 3, E is not formally defined, and both  |E|and e are used for the number of environments.
> In Section 3, π (softmax?) is used without its definition
> In Equation (1). σc must be αc.
> At the end of Page 6, Table 15 must be Table 5.
>
> Thank you for this valuable feedback. We will update the paper accordingly.

---

> > ### Comment · Reviewer_TAie · 2024-11-20
> >
> > > Does the environment-specific variation include "negative" variation to counteract the global "top" words in some environments?
> >
> > >>  Can you elaborate on what you mean? If our reading of this point is correct, we think that our model already does this. In our model, if a word does not appear in an environment, it is less likely to be considered in the global topic beta words.
> >
> > I just wanted to confirm that the environment-specific weight gamma can be negative.
> > In Tables 5 and 6, I believe the "top words" in "$\gamma_k$: Republican", for example, mean the words with large positive gamma weights. It may also be informative to show the words with large negative gamma weights if it is relevant.

---

> > > ### Author Response · Authors · 2024-11-20
> > >
> > > >I just wanted to confirm that the environment-specific weight gamma can be negative.
> > >
> > > Yes, they can be negative.
> > >
> > > >It may also be informative to show the words with large negative gamma weights if it is relevant.
> > >
> > > This is an interesting idea, thanks for sharing.

---

### Official Review · Reviewer_rSS2 · 2024-11-03

**Soundness:** 3
**Presentation:** 3
**Contribution:** 2
**Rating:** 3
**Confidence:** 4

**Summary:**

This paper introduces Multi-Environment Topic Models (MTM) as a novel addition to unsupervised topic modeling, designed to differentiate between global and environment-specific patterns in text data–a capability that traditional models lack. MTM leverages mean-field variational inference to approximate the posterior distribution and introduce a new latent variable, $\gamma$, to isolate environment-specific information. As a hierarchical model, it employs Automatic Relevance Determination (ARD) as a prior to capture environment-based variations while preserving interpretability. MTM demonstrates stable predictive performance across diverse environments in both in-domain and out-of-domain settings and shows enhanced accuracy in causal inference tasks, particularly where distinguishing the impact of environment-specific language is essential.

**Strengths:**

MTM is a novel addition to the family of topic models, offering a generative model that introduces a global topic-environment-word distribution, $\gamma$, to enhance adaptability across multiple environments. By using an ARD prior to enforce sparsity, MTM is carefully designed to align with the paper’s goal of interpretable, environment-specific topic modeling. The paper presents two variations, MTM and nMTM, demonstrating flexibility in approach and an attention to design that supports robust generalization. Experiments effectively highlight MTM’s capabilities, particularly its utility in applications requiring causal accuracy.

Overall, the paper is well-structured and provides a clear, detailed description of the MTM methodology and its motivation. The thorough explanation of each experimental setup, combined with baseline comparisons, enhances understanding of MTM’s strengths in multi-environment contexts. Additionally, figures and tables effectively illustrate MTM’s performance, contributing to the paper’s clarity and readability.

**Weaknesses:**

- Some minor typos:
    - line 266 "by by"
    - line 443 "r\ $\hat{\theta_i}$ \ ${\hat\theta}_i$"

- It would be beneficial to include a more thorough analysis of sparsity enforcement and the selection of the ARD prior versus other priors. Providing ablation studies or theoretical comparisons with alternatives (e.g., the Horseshoe prior) would strengthen the rationale behind this choice and clarify its role in enhancing MTM’s performance, as suggested by the evaluation metrics.
- Expanding the experiments beyond political and media datasets could illustrate MTM’s versatility and applicability. Datasets from scientific literature or product reviews, for example, could demonstrate the model’s adaptability and provide insights into its generalization capabilities across domains.
- While MTM generally outperforms its baselines, a more detailed exploration of MTM and nMTM would be valuable. Discussing their respective strengths and limitations, along with situations where one might be preferred, could help users understand these variations better.
- Although known techniques are employed, incorporating a detailed derivation of the ELBO alongside the algorithm would be beneficial. Topic models are widely used outside of machine learning, and a clearer explanation could make the paper accessible to a broader audience.
- The presented generative process and inference approach is novel. However, the family of topic models is extensive, and alternative baselines might provide a more comprehensive comparison. For instance, MixEHR [1,2] uses CVB and could treat each specialist type as an environment, could serve as an interesting baseline for MTM, and highlight the model’s strengths.

[1] Song, Z., Toral, X. S., Xu, Y., Liu, A., Guo, L., Powell, G., ... & Li, Y. (2021, August). Supervised multi-specialist topic model with applications on large-scale electronic health record data. In Proceedings of the 12th ACM Conference on Bioinformatics, Computational Biology, and Health Informatics (pp. 1-26).

[2] Li, Y., Nair, P., Lu, X. H., Wen, Z., Wang, Y., Dehaghi, A. A. K., ... & Kellis, M. (2020). Inferring multimodal latent topics from electronic health records. Nature communications, 11(1), 2536.

**Questions:**

I am open to revisiting my score if the authors address the points outlined in the weaknesses and provide clarification.

---

> ### Author Response · Authors · 2024-11-19
>
> >It would be beneficial to include a more thorough analysis of sparsity enforcement and the selection of the ARD prior versus other priors. Providing ablation studies or theoretical comparisons with alternatives (e.g., the Horseshoe prior) would strengthen the rationale behind this choice and clarify its role in enhancing MTM’s performance, as suggested by the evaluation metrics.
>
> Section E in the appendix provides a detailed comparison of the MTM, horseshoe MTM, and nMTM. We prefer the MTM because it is better than the baselines at identifying terms that are unique to a particular while also being more sparse.
>
> >Although known techniques are employed, incorporating a detailed derivation of the ELBO alongside the algorithm would be beneficial. Topic models are widely used outside of machine learning, and a clearer explanation could make the paper accessible to a broader audience.
>
> Thanks for the suggestion. We will add this derivation to the appendix.
>
> >Expanding the experiments beyond political and media datasets could illustrate MTM’s versatility and applicability. Datasets from scientific literature or product reviews, for example, could demonstrate the model’s adaptability and provide insights into its generalization capabilities across domains.
>
> That’s an interesting idea! What is the use-case you envision? Our focus is on political datasets because they are commonly used by social scientists to estimate causal effects.
>
> >The presented generative process and inference approach is novel. However, the family of topic models is extensive, and alternative baselines might provide a more comprehensive comparison. For instance, MixEHR [1,2] uses CVB and could treat each specialist type as an environment, could serve as an interesting baseline for MTM, and highlight the model’s strengths.
>
> These are interesting papers. MixEHR has a different goal from the MTM. It focuses on modeling the heterogeneity of each patient, while the MTM aims to learn a global representation while capturing the environment specific effects on the global topic representation.

---

### Official Review · Reviewer_MNG1 · 2024-11-05

**Soundness:** 3
**Presentation:** 2
**Contribution:** 2
**Rating:** 3
**Confidence:** 4

**Summary:**

Topic modelling is an interesting research area that has been explored for over a decade, evolving from probabilistic models to neural topic models. This paper introduces a probabilistic generative model designed to model documents generated from various contexts, or "environments." Unlike the traditional topic model, such as Latent Dirichlet Allocation (LDA), this model incorporates environment-specific topic-word distributions. It assumes that, in addition to words, each document is associated with an environment indicator variable. Variational inference is then used to learn the model parameters. The proposed model is evaluated against several baseline topic models across three datasets, with metrics including perplexity, topic coherence, and causal estimation.

**Strengths:**

* The introduction of environment-specific topic-word distributions is interesting, which can be seen as one strength of this paper in terms of modelling. The final categorical distribution used to generate each word combines the shared per-topic word distribution, denoted as $\beta_k$, with the environment-specific per-topic word distribution represented by $\gamma_{e, z}$.

* The application of the proposed model in causal estimation adds another interesting dimension to this work. Based on the potential outcome framework, it computes the Average Treatment Effect (ATE). In this causal estimation process, the words in a document are treated as covariates that influence the outcome when the document is processed within a specific environment. The experimental results derived from the synthetic dataset show that the proposed model can capture the effect caused by adding 0.2 to the outcome variable.

**Weaknesses:**

* Modeling different environments or contexts through environment-specific topic-by-word matrices resembles the approach of using hierarchical structures to model various types of document collections. For instance, the paper on "Differential Topic Models" employs a hierarchical Pitman-Yor Process to model and compare different document collections, while "ContraVis: Contrastive and Visual Topic Modeling for Comparing Document Collections" also enables the comparison of distinct document collections, among others. Therefore, the technical contribution of this work appears somewhat limited.
* It is not surprising to see the perplexity improvement, given that the proposed model makes use of extra supervised information, i.e., the environment indicator variable.
* There is a lack of comparison with those models that are capable of modelling different corpora, like these two models mentioned above.

**Questions:**

* Line 158: should $z$ be $z_{i,j}$?
* Lines 157 and 158: what does $\pi()$ mean?
* Eq (1): Should $\sigma_c$ be $\alpha_c$?

---

> ### Author Response · Authors · 2024-11-19
>
> > Differential Topic Models
>
> We are interested in learning a global topic distribution; differential topic models are good at finding topics that vary across different data collections but do not contribute to our goal of learning a unified topic distribution for each topic that can be used in downstream causal experiments. DTMs share some similarities, but are relatively limited compared to other metadata models such as the structural topic model or SCHOLAR, which we include. Using DTMs instead of MTM would lead to unbiased causal estimates in our setting, which is motivated by real social science use-cases.
>
> > ContraVis: Contrastive and Visual Topic Modeling for Comparing Document Collections
>
> ContraVis classifies entire topics into local or global categories, allowing for contrastive analysis of differences and similarities between document collections. In contrast, the MTM learns a unified, global version of each topic, with a parameter representing the environment-specific adjustments on that global distribution. This global topic distribution in MTM supports causal inference by enabling consistent topic proportions for documents across different environments, rather than each document from different environments having different topics assigned to it.
>
> > It is not surprising to see the perplexity improvement, given that the proposed model makes use of extra supervised information, i.e., the environment indicator variable.
>
> More information can lead to perplexity to improvements; however, sparsity plays a more important role in improving model performance. The nMTM includes a non-sparse prior and consistently performs worse than the MTM, and even worse than the VTM on some experiments despite the VTM containing no environment specific information.
>
> >There is a lack of comparison with those models that are capable of modelling different corpora, like these two models mentioned above.
>
> The SCHOLAR is an example of a model that is widely used by social scientists that models environment metadata.
>
> >Questions
>
> Thank you for your questions. π() represents the softmax and σc should be αc in eq.1.

---

### Meta-Review · Area_Chair_5qRF · 2024-12-20

**Metareview:**

This paper concerns the development of Topic Models. This paper proposes a multi-environment topic model which can discover global and environment specific topics. This paper proposes an unsupervised model, in line with LDA and shows superior performance against baselines.
Given that topic models have been developed significantly this paper looks a little preliminary, as evidenced by the reviewer feedback. The author(s) should consider developing this line.

**Additional Comments On Reviewer Discussion:**

The reviewers remained unconvinced that the current contribution can be considered strong enough for ICLR.

---

### Decision · Program_Chairs · 2025-01-22

Reject